# Drugs That Changed Society: Microtubule-Targeting Agents Belonging to Taxanoids, Macrolides and Non-Ribosomal Peptides

**DOI:** 10.3390/molecules27175648

**Published:** 2022-09-01

**Authors:** Søren Brøgger Christensen

**Affiliations:** The Museum of Natural Medicine & The Pharmacognostic Collection, University of Copenhagen, DK-2100 Copenhagen, Denmark; soren.christensen@sund.ku.dk; Tel.: +45-3533-6253

**Keywords:** paclitaxel, docetaxel, cabazitaxel, eribulin, colchicine, podophyllotoxin, Halochondrin, taccalonolide, Zamopanolide, microtubule-targeting agent

## Abstract

During a screening performed by the National Cancer Institute in the 1960s, the terpenoid paclitaxel was discovered. Paclitaxel expanded the treatment options for breast, lung, prostate and ovarian cancer. Paclitaxel is only present in minute amounts in the bark of *Taxia brevifolia*. A sustainable supply was ensured with a culture developed from *Taxus chinensis,* or with semi-synthesis from other taxanes. Paclitaxel is marketed under the name Taxol. An intermediate from the semi-synthesis docetaxel is also used as a drug and marketed as Taxotere. O-Methylated docetaxel is used for treatment of some paclitaxel-resistant cancer forms as cabazitaxel. The solubility problems of paclitaxel have been overcome by formulation of a nanoparticle albumin-bound paclitaxel (NAB-paclitaxel, Abraxane). The mechanism of action is affinity towards microtubules, which prevents proliferation and consequently the drug would be expected primarily to be active towards cancer cells proliferating faster than benign cells. The activity against slowly growing tumors such as solid tumors suggests that other effects such as oncogenic signaling or cellular trafficking are involved. In addition to terpenoids, recently discovered microtubule-targeting polyketide macrolides and non-ribosomal peptides have been discovered and marketed as drugs. The revolutionary improvements for treatment of cancer diseases targeting microtubules have led to an intensive search for other compounds with the same target. Several polyketide macrolides, terpenoids and non-ribosomal peptides have been investigated and a few marketed.

## 1. Introduction

Singer reports a dramatic increase in cancer mortality in the period 1900 to 1950 in England and Wales [1]. In the same period, a dramatic decrease is observed in the death rate of tuberculosis and other infectious diseases. Most likely the increased death rate for cancer is caused by an increase in life expectancy since cancer is a disease most frequently appearing in populations of elderly persons. Recent statistics confirm that cancer is the main reason for death among older people in countries with a high human development index [1], where death rates of communicative diseases are almost negligible [2,3]. Among men, lung cancer, colorectum cancer and liver cancer are the main cause of death [2]. Among women, causalities caused by lung, breast and colorectum cancer dominate in countries with high human development index. Breast, cervix uteri and ovary cancer dominate causes of death in countries with medium human development index [2]. In the Nordic countries (Denmark, Faroe Islands, Finland, Greenland, Norway and Sweden with a population of 28 million), the annual incidence of cancer cases is about 98,000 among males and 82,000 among females and mortality is about 34,000 in males and 30,000 in females. The incidence rate is increasing, whereas mortality is decreasing. The incidence for breast cancer is 21,000 for females and mortality 4000 per year. For lung cancer, the incidence is 8000 for males and 7500 for females and the mortality for males is 6400 and 5500 for females per year. For colorectum cancer, the incidence is 10,700 for males and 9500 for females; mortality is 3800 for males and 3400 for females per year [4]. The difference between incidence and mortality reveals that increasing numbers of patients are living with a cancer disease. The number of cancer patients living with cancer reveals that modern drugs enable patients to live in periods without progression of the disease and some can even cure the disease. The situation calls for improved drugs for long-term treatment or combatting cancer. The program for natural products of the National Cancer Institute headed by Dr. Hartwell afforded camptothecins, the maytansine antibody conjugate and drugs derived from the plant belonging to the genus *Taxus.* These drugs have improved progression-free survival for patients suffering from several cancer diseases such as breast, ovarian, non-small cell lung cancer and Kaposi’s sarcoma [5,6,7,8]. Paclitaxel has been the best-selling chemotherapeutic in history with sales of over USD 1 billion in 1998 [9] and sales of USD 1.6 billion in 2000 [10]. In recent years, new drugs have to some extent replaced paclitaxel.

The present review emphasizes the chemistry of the taxanoids and the clinical progressions obtained by introduction of new semisynthetic analogues. New sources for natural products have led to macrolides and peptides which may improve the possibilities for treatment of cancer diseases [11].

## 2. The *Taxus* Diterpenes

### 2.1. The Genus Taxus

The genus *Taxus* belong to the plant family Taxaceae. The absence of cones initially excluded Taxaceae from the Gymnospermae (conifers) but later botanists included the family within this division of the plant kingdom [12,13]. The genus *Taxus* comprises according to Plant of the World nine species *T. baccata* L., *T. brevifolia* Nutt., *T. canadensis* Marshall, *T. chinensi* (Pilg.), *T. contorta* Griff. (synonym *T. fauna* Nan Li & R.R.Mill), *T. floridana* Nutt. ex Chapm., *T. globosa* Schltdl., *T. cuspidata* Siebold & Zucc., *T. mairei* (Lemée & H. Lév.) S.Y.Hu and *T. wallichiana* Zucc. [14]. Species belonging to the genus are found in temperate America and Eurasia. *T. baccata* L. (English yew) is endemic to Britain. Longbows made from this tree were terrifying weapons used by the English against the French during the Hundred Years War [15]. The toxic effects of yew extracts have been known since ancient times. Celts are reported to have used *T. baccata* extracts to poison their arrows and have used the poison to commit ritual suicide [16]. Dioscorides, Pliny the Elder, Galen and Julius Cesar also mention accidental deaths caused by the poison [13]. Not only are the poisonous properties of *Taxus* reported in the scientific literature but also Shakespeare in his play “Macbeth” and Agatha Christie in a “Pocket Full of Rye” take advantage of this property. Native Americans used extracts of *T. brevifolia* as disinfectant, abortifacient or for treatment of skin cancer [14], even though it might be dubious if they could make a correct diagnosis. Drugs made from *T. baccata* are used in Kashmir for treatment of cancer and tumors, but no specification of the cancer disease is given [17]. Himalayan yew (*T. wallichiana*) has been used for treatment of headaches and snake bites in southeast and central China [14]. In Nepal, the leaf juices of *T. wallichiana* and *T. contoria* have been used for treatment of cough, fever, gastro-intestinal problems and cancer [16]. Again, it might be questioned how the cancer has been diagnosed. In France, an aqueous extract of the leaves of *T. baccata* has been used against rheumatism. Minute amounts of the main active principle prevented isolation and structure elucidation for some years. The compound (Figure 1, **1**) was given the trivial name paclitaxel. Taxol is a trade name for paclitaxel.

### 2.2. Taxanes Isolated from the Taxus Genus

A taxane is a diterpenoid isolated from a plant belonging to the genus *Taxus*. Another nomenclature system claims that only diterpenoids belonging to groups I and Ia (Figure 2) should be named taxanes [13]. An early report on compounds isolated from *T. baccata* describes how a fraction named taxin is obtained by extraction of an acidic aqueous solution after addition of ammonia with organic solvent. No toxicity investigation was performed on this product. A later study determined a molecular formula of C_37_H_52_NO_10_ [18]. Several grams of taxin were isolated [19,20]. The isolation procedure involved partitioning between acidic and basic media as is usual for alkaloids. This procedure, however, precludes paclitaxel from being present in the isolated amorphous powder. Paclitaxel is an amide (1) and consequently does not possess basic properties. Moreover, the amount of product isolated, several grams, is suspicious since paclitaxel is only present in minute amounts in the bark and leaves. An isolated yield of paclitaxel from one tree varies between 90 mg and 700 mg [21]. Later studies revealed that taxin was a mixture of compounds. Acidic hydrolysis of the product afforded 3-dimethylamino-3-phenylpropionic acid (Figure 1, **2**) [18] and four other phenylisoserine derivatives (**3**–**6**). The presence of an amino group in the side chain makes some taxanes pseudoalkaloids. A requirement for an alkaloid is the presence of a residue of an amino acid in the alkaloid skeleton [22]. The presence of a hydroxy group in the α-position of the side chain as in 3 is important for affinity for the binding site in tubulin and consequently for the cytotoxicity. About 45 esters of diterpenes esterified with amino acid are found in *Taxus* species. In *T. baccata*, about 12 are found [14]. Consequently, it is reasonable to conclude that taxins other than paclitaxel are major constituents in the fraction named taxin [23].

The nomenclature is confusing. In early reports, the natural product is reported as taxin [18,19]. Since the compounds are pseudoalkaloids, the suffix should be “ine”, the general accepted suffix for alkaloids and pseudoalkaloids [22,24].

Several esterified diterpenoids with different skeletons have been isolated from *Taxus* species. A recent review estimates the number to be 600 [23]. Some carbon skeletons are shown in (Figure 2). The isoprene rule does not apply to groups III and V, indicating that a rearrangement has occurred during the biosynthesis of these skeletons. Groups VI and X are a seco skeletons because a ring has been opened. The isoprene rule only applies to group X unless cleavage of a ring is considered.

In the different skeletons C-1, C-2 and C-7 are most frequently unsubstituted, but occasionally substituted with a hydroxy group, which may be acylated. C-3, C-5 and C-10 are almost always hydroxylated. C-9 might be a secondary alcohol or a ketone. C-13 is almost always hydroxylated and very frequently the hydroxy group is acylated. C-14 might be hydroxylated. The C-4-C-20 bond is a double bond in many cases, but examples of epoxides are found. Some typical diterpenes are shown in Figure 2. The dominating acid acylating the alcohol groups is acetic acid. Some other aliphatic acids such as 2-methyl-butanoic acid, 3-hydroxy-2-methyl-butanoic acid 3-hydroxy-2-methyl butanoic acid, 3-hydroxybutyric acid, hydroxyacetic acid, acetoxyacetic acid, tiglic acid, hexanoic acid, heptanoic acid, 4-methylheptanoic acid, octanoic acid, cis-dec-9-enoic acid and isobutyric acid have also been found. Among the aromatic carboxylic acids, benzoic and cinnamic acid are the most frequent, but cis-3-phenylpropenoic acid, 2-acetoxycinnamic acid, 3-dimethylamino-3-phenylpropionic acid, 3-methylamino-3-phenylpropionic acid and 3-dimethylamino-2-hydroxy-3-phenylpropionic acid have also been found. The presence of some amino acids affords some compounds basic properties [22]. From a clinical point of view the most interesting taxanes are acylated with 3-amino-3-phenylpropionic acid and 3-amino-2-hydroxy-3-phenylpropionic acid (isophenylserine) N-acylated with benzoic acid, formic acid, propionic acid, butanoic acid, tiglic acid, cinnamic acid, hexanoic acid, heptanoic acid, 4-methylhexanoic acid, 2-methylbutanoic acid, octanoic acid and cis-dec-7-enoic acid. In addition, in order to be clinically interesting, the compounds must possess an oxetane ring [9].

### 2.3. Other Sources for Paclitaxel

The poor yield of paclitaxel from the bark of the yew has inspired dereplication to find other organisms producing the compound. Several endophytes have been investigated. Paclitaxel has been found in the broth of the endophyte *Taxomyces andreanae* isolated from *T. brevifolia* [25,26]. About 150 different endophytes have been isolated from *T. baccata* and 10% of these have the potential to produce paclitaxel. Unfortunately, they frequently lose this ability by subculturing [27]. Cultivation of *T. andreannae* afforded paclitaxel but not in commercially interesting amounts [28]. The occurrence of paclitaxel synthesis in *Taxus* species and in their endophytes suggests that gene transfer has taken place. Phylogenic analyses suggest no recent gene transfer between plants and fungi [29]. Surprisingly, paclitaxel (**1**), 10-deacetyl baccatin III (**8**), bacatin III (**8**) and cephalomannine (**21**) have been discovered in the shells, leaves and green cover shells of *Corylus avellana* L (Betulaceae, Tombul hazelnut), but only in trace amounts [25]. In addition, paclitaxel has been found in *Afroocarpus gracilior* Pilger C.N.Page (synonym of *Podocarpu graciliar*) (Podocarpaceae) [30] and in *Cephalotaxus hainanensis* H.L.Li (Taxaceae) [31].

### 2.4. Isolation and Structure Elucidation of Taxol and Other Taxanes

#### 2.4.1. Structure Elucidation of Taxanes

The first alkaloidal fraction isolated in 1856 from *Taxus* species was named taxin [32]. Later studies revealed that the product was a complex mixture of at least eleven pseudoalkaloids [33,34]. It was not until a century after the isolation of taxin that the constitutions of the compounds were elucidated. Taxine B together with some analogues were obtained in a pure state in gram quantities by counter-current chromatography [33]. Based on the chemistry of the molecule, it was concluded that taxine B consists of a diterpene (C_20_H_25_) skeleton substituted with 3-dimethylamino-3-phenylpropionyloxy (C_11_H_14_NO_2_), acetoxy (C_2_H_3_O_2_), ketone (O) and three hydroxy groups [(HO)_3_] [33,34]. A later investigation based on 2D NMR studies revealed the acetylation pattern and finally established the structure 7 (Figure 3) [35]. Taxine A, which was another pseudoalkaloid present in taxine, possesses the structure 10 (Figure 3) [13,14].

The structure of taxine I (**15**) was suggested based on isolation of compound 16 (Figure 4). Since 15 appeared in a fraction obtained after partitioning between acidic water and organic phases, it was assumed that treatment with base had deaminated the isoserine unit during the isolation [36]. In 1968, the paradoxical situation was described as “the constituents of taxin which have been isolated (taxine A, B and C) have not been structure elucidated and that which has not been isolated (taxine I) has been structure elucidated” [36]. The compound was later isolated from the seeds of *T. baccatus* and named diacetyltaxine B (**16**) [35,37]. The name taxine I is no longer used in the literature [14,23].

#### 2.4.2. Structure Elucidation of Paclitaxel

A bioguided isolation led to an amorphous cytotoxic compound, the structure of which was elucidated with methanolysis and X-ray analysis of the 10,13-diiodoacetate ester of the terpenoid alcohol (Figure 5). The experimental conditions for preparing the diiodoacetate are not reported [38]. The previous study on constituents isolated from *Taxus* species included partitioning between acidic and basic media. The acylation of the amine group in paclitaxel prevents it from being present in the products obtained after partitioning between acidic and basic media.

#### 2.4.3. Biosynthesis of Taxanes

The biosynthesis of paclitaxel starting from geranylgeranyl pyrophosphate is depicted in Figure 6 [23]. Many of the enzymes involved in this reaction path are poorly characterized. The introduction of the oxetane groups is assumed to proceed after epoxidation of the C-4-C-20 double bond [39]. Oxidation and benzoylation of the isophenylserine is assumed to take place after esterification with the C-13-hydroxy group of the taxane [39]. All the oxidative reaction steps are catalyzed by cytochrome P450 oxygenases [39]. The exact sequence and some steps like the oxetane formation are still debated [29].

An acid-catalyzed mechanism for the formation of oxetane is presented in Figure 7 [40,41]. The proposal explains the stereochemistry of the formed oxetane ring but includes an S_N_2 attack on the quaternary C-5. Two non-acid-catalyzed mechanisms have also been suggested [41].

#### 2.4.4. Chemistry of Taxanes

Paclitaxel (**1**) only occurs in minute amounts in the bark of the slow growing tree *T. brevifolia.* About 1 g of paclitaxel can be isolated from the bark of three trees. Assuming that an average patient is dosed with 2.5 g for a treatment, then 400,000 patients would need 1000 kg of paclitaxel for a treatment [22,26]. The annual demand is assumed to be 1000 kg, revealing that either the compound must be isolated from cell cultures, or the compound must be obtained by semi-synthesis from other taxanes. According to the homepage of Phyton Biotech Gmbh, they can supply the world with paclitaxel produced in an environmentally friendly manner by plant cell fermentation using a strain developed from *T. chinensis* v. *marei* [42]. Some cell cultures of *Taxus* sp. and endophytes have been screened for their ability to produce paclitaxel. Cultures of *Aspergillus fumigatus* isolated from *Taxus* sp. of the Northern Hamalayan region are reported to produce 1.6 paclitaxel per liter medium. Unfortunately, the purity of the product appears to be very low [43]. Alternative sources of paclitaxel could be other organisms producing metabolites, which can be used as starting material for semisyntheses. This approach has been used for sustainable supply of the payload of mipsagargin [44] or for preparing artesunate and artemether from artemisinin [45]. Taxadiene (the first tricyclic compound in Figure 6) has been produced in a cell culture of modified *Escherichia coli* in a yield of 1 g per liter [46]. No simple procedure for converting taxadiene to paclitaxel exists. The almost inseparable mixture of cephalomannine (Figure 8, **21**) and paclitaxel (**1**) obtained by purifying paclitaxel from an extract of bark of *T. brevifolia* [47] is another option. By semisynthesis, this mixture can be converted into pure paclitaxel (Figure 8). The first step is a de-esterification taking advantage of the α-hydroxy group in the O-13 substituent. A borohydride complex with this hydroxy group enables a selective reduction of this carbonyl group to give a hemiacetal, which is easily hydrolyzed to give **24** and the two diols **22** and **23** [47].

Reacetylation of **24** requires use of a protecting group since the 7-hydroxy group reacts faster than the 13-hydroxy group. A procedure involving masking of the 7-hydroxy group followed by acylation of the 13-hydroxy group and demasking has been developed (Figure 9, **27**) [47].

The 2′-hydroxy group of paclitaxel (**1**) is more reactive than the 7-hydroxy group, meaning that selective acylation of the 7-hydoxygroup can only be performed if the 2′-hydrxy group first is protected. After deprotection of the 2′-hydroxy group, the 7-O-acylted derivative is obtained. The tricholoroethylcarbonyl group (Troc) has been used for protecting the 2′-hydroxy group [48]. The Troc group is seldom used as the protective group. However, it is used in taxane and carbohydrate chemistry [48,49].

Paclitaxel (**1**) can also be obtained by a semi-synthetic approach using 10-deactyl baccatin III (**28**), which can be isolated from the needles of *T. baccata* in a yield of 1 mg/kg [50]. The tree regenerates the needles, enabling repeated prudent harvesting to give large amounts of **28** (Figure 10). Since, as shown in Figure 9, 7-OH is acylated far faster than O-10 and O-13, masking of this hydroxy group to give **17** is needed to enable selective acetylation of O-10. 4-Dimethylaminoopyridine (DMAP)-catalyzed acylation of **30** with phenylisoserine in which the hydroxy group is protected as an acetal (**31**) successfully afforded O-13 acylation to give **32**. The carboxylic acid is activated by di-2-pyridyl carbonate (DPC). Demasking with hydrochloric acid afforded paclitaxel (**1**) [50].

The sterically hindered 13-OH group of baccatin III (**28**) is difficult to acylate with bulky isoserine derivatives such as **31**. Instead, the cinnamoyl derivative **33** (Figure 11) easily prepared from protected baccatin III (**33**) has been suggested as a starting material paclitaxel synthesis. (**28**) [51]. As can be seen in Figure 11, a Sharpless oxyamination affords a mixture of all possible regio- and stereoisomers **34a(**2′R,3′S)/**34b**(2′S,3′R) and **36a(**2′R,3′S)/**36b**(2′S,3′R). Compounds **34a**/**34b**, **35a**/**35b, 36a**/**36b** and **37a**/**37b** are mixtures of the two diastereomeric compounds as illustrated for **38a** and **38b**. Attempts to increase enantiomeric excess using quinine alkaloids as chiral catalysts failed [13,51]. Compound **34a** is obtained during this reaction sequence. Removal of the Troc groups using zinc in acetic acid from the mixture of the four isomers **34a/34b** and **36a/36b** afforded four products. A Boc group was introduced in one of these to give docetaxel (**39**). The trade name for docetaxel is Taxotere. This agent is more efficient in inhibiting disassembly of microtubules than paclitaxel and is used in the clinic [13]. Another advantage of docetaxel is higher solubility in aqueous media.

The use of β-lactames for introduction of the 3-amino-2-hydroxy-3-phenypropionic acid was first described in a patent [52]. β-lactame **43** is obtained in racemic form by reacting acetoxyacetyl chlorid (**40**) with the imine obtained by reacting benzaldehyde with 4-methoxyaniline ((Figure 12, **41**). The 4-methoyxphenyl group is oxidative removed (**43**) and the acetoxy group was converted to a hydroxy group masked as ethoxyethyl acetal to give **44**. Products **42** and **43** are racemic mixtures. The number of isomers in products **44**, **45** and **46** is further increased by the presence of a chiral protecting group. After 4-dimethylaminopyridine catalyzed reaction between **45** and 7-O-trimethylsilylbaccatin III (**46**), paclitaxel masked at O-2′ and 7-O is obtained (**47**) as a mixture of isomers (2′R, 3′S and 2′S,3′R). The complicated stereochemistry caused by the chiral-protecting group is resolved by demasking with hydrochloric acid to give paclitaxel (**1**) and the diasteromer with opposite stereochemistry of the two stereogenic carbon atoms 2′ and 3′ in the isophenylserine group (Figure 12) [52].

A procedure for selective methylation of O-7 and O-10 has been developed for deacetylbaccatin III [53] (Figure 13, **28**). Acylation of **48** using β-lactam **50** affords cabazitaxel (**49**) after deprotection. Several patents for efficient O-13 esterification of masked deacetylbaccatin III like **48** using either the oxazolidine (Figure 14) or β-lactame approach (Figure 12 and Figure 13) have been approved [54,55,56,57,58].

The use of 2-hydroxy-3-amino-carboxylic acid masked as an oxazolidine (**55**) to esterify O-13 at **28** is illustrated in Figure 14 [59].

### 2.5. Mechanism of Action of Paclitaxel

Microtubules are filamentous, tube-shaped protein polymers. They are essential for maintaining cell shape, transport of proteins and organelles such as mitochondria and vesicles. They also are essential in cell signaling [60]. They also play an important role in cell migration, maintaining cell shape and polarity and compartmentalizing. The microtubule network is essential for maintaining most cellular functions and cannot be replaced by other cellular functions [60]. In addition, they play an essential role in cell division [5]. Microtubules consist of polymers of α- and β-tubulin heterodimers in the form of filamentous tubes (Figure 15). The functions of microtubules are regulated through binding of regulatory proteins including microtubule-associated proteins (MAP) [5]. The formation of the microtubule is initiated by slow formation of a nucleus followed by a fast elongation at the ends by α,β-tubulin dimers. Once formed, the microtubule continues to switch between assembly and disassembly (dynamic instability). The two ends are not identical; the plus end grows and shortens faster than the minus end. In treadmilling, the microtubule shortens in one end and elongates in the other end, causing an internal flow of tubule dimers from one end to the other. The minus end is commonly anchored at the centrosome-containing microtubule-organizing center (MTOC) adjacent to the cell nucleus (Figure 16) [60]. An important function might be trafficking of essential enzymes, which is blocked by microtube-targeting agents [60].

During mitogenesis (Figure 16), the microtubules are arranged in a spindle attached to kinetochores, enabling them to perform the alignment at the metaphase plate (congression). During the prometaphase, the nuclear envelope degrades, and the chromosomes are congressed to the equator of the cell. In the anaphase, the two DNA strings of chromosomes attached to the microtubules are moved towards the spindle poles (Figure 16). In the telophase, the cell is dividing into two daughter cells [5].

The interest for paclitaxel as a drug candidate intensified when it was discovered that it promoted assemblance of α- and β-tubulin microtubules, and dissemblance was hindered [61]. Whereas paclitaxel was the first compound discovered to promote tubulin polymerization, several examples of inhibitors of tubulin assemblance were known such as the vinca alkaloids, vinblastine and vincristine, combretastatin, halichondrin, eribulin, dolastatin, noscapine, hemiastertin, colchicine and podophyllotoxin (Section 3) [5,62,63].

It has generally been assumed that microtubule-targeting agents disturb the mitosis by interfering with the microtubules during the mitotic process. In vitro studies incubating cancer cells proliferating with a rate of 24 h with antimitotics results in aberrant formation of cells in the G_2_/M phase within 24 h. Eventually, this leads to apoptosis [21,60]. However, solid tumors in patients have a rate of cell division ranging from 150 to 300 days, which is longer than many benign cells [60]. Consequently, microtubule-targeting agents should be just as toxic towards benign cells as cancer cells, unless other mechanisms are involved. Microtubules in nondividing cells are important for cellular signaling both for intracellular trafficking and for scaffolds facilitating protein–protein interactions [60,64]. Cancer cells depend on oncogenic-signaling pathways that rely on functional microtubules. These pathways may be disturbed by antimitotics [60]. Paclitaxel also increases reactive oxygen species (ROS) and an overexpression of genes and proteins related to stress. Damage to the membrane of the endoplasmic reticulum may cause Ca^2+^ release, provoking damage of the mitochondria [6]. Several other pathways, including the TLR-4 cascade and the NLRP3 inflammasone, have been suggested for the mechanism of action of paclitaxel [6]. Further evidence for a more complicated mechanism of action is illustrated by the cytotoxicity of some D-seco analogues of paclitaxel. A number of these have a 100-fold to 1000-fold higher binding affinity for microtubules than paclitaxel but possess a 10,000-fold smaller cytotoxicity [65].

A major drawback of paclitaxel is the development of resistance. Reduction of tumor-suppressing genes may be involved in development of paclitaxel resistance. Proteins such as hypoxia factor 1 (HIF-1) and keratin 17 have also been suggested to be involved in development of resistance [6]. Multidrug resistance-associated protein 1(MDR1) and P-gp are pumps that can remove paclitaxel from the cell. Inhibitors of these pumps prevent efflux from the cell [6]. A number of other proteins have been correlated with paclitaxel resistance, including forkhead protein 1 and the receptor tyrosine kinase inhibitor. Moreover, the expression of different α and β tubulin isotypes also affects the sensitivity of the cells towards paclitaxel. At present, however, the topic of paclitaxel resistance is poorly understood [6].

### 2.6. Structure Activity Relationships of Paclitaxel

Paclitaxel (Figure 1, **1**) binds at the 1–31 and 217–233 amino acid residues of the β-tublin subunit, at the inner surface of the microtubule lumen [21]. It has a critical hydrophobic interaction with His229 in the binding pocket of β-tubulin with the C-3′-benzamido group, and two critical hydrogen bonding processes from the NH-backbone of Arg369 to the C-2′-OH group and from Thr276 to the oxetane oxygen [66]. Moreover, the carboxyl-OH of Asp26 and the backbone NH of Gly370 are at a distance from C-2′-OH enabling hydrogen bond formation [67]. Accordingly, analogues missing the C-2′-OH (Figure 17, **60**) group have poor affinity for the binding site [48,67]. Acylation at O-2, O-4 and N-3′ is essential for activity [21]. Concerning the C-3′-benzamido group, analogues possessing other hydrophobic acyl groups such as *t*-butoxycarbonyl (docetaxel, **39**) also have high affinity [67]. It is suggested that the primary effect of the C-3′-substituent is to twist the sidechain, enabling hydrogen bonds for C-2′-OH [67]. Several analogues in which the benzamide group and the C-3′ substituents were replaced with other groups have been patented. Analogues in which the groups at C-2, O-7, O-9 and O-10 were replaced were also prepared. Even though bioassays were performed, the structure activity relationships are difficult to understand [56,57]. In accordance with the considerations concerning the O-13 side chain baccatin III (**8**), those with no sidechain at O-13 and 2′-dexoxypaclitaxel (**60**) are 1000 times or 100 times, respectively, less potent than paclitaxel [67]. Structure activity relationships such as the importance of the stereochemistry at C-2′ and C-3′, the presence of an acyl group at N-3′ and the presence of a hydroxy group at C-2′ for high affinity all confirm the suggested pharmacophore [66,68].

The affinity towards β-tubulin is less sensitive to structural change in other parts of the molecule. Changes in the C-7 to C-12 moiety such as change of the stereochemistry at C-7, acetylation of O-7 or deacetylation of O-10 only reduce the affinity to a small extent. 7-O-xyloside and 7-O-glutaroyl are equipotent to paclitaxel. Esterification of O-7 with N,N-dimethyl glycine or alanine gives compounds with half the affinity of paclitaxel (**1**) [68]. The C-1 to C-6 part of the molecule is less investigated because of difficult chemistry. Exchange of the O-2 benzoate with substituted benzoic acids such as m-hydroxybenzoate did not change the affinity to β-tubulin, but made the compound less cytotoxic [68]. Replacement of oxygen in the oxetane ring with nitrogen to obtain an azetidine (Figure 18, **61**) reduces the affinity eight times [69]. This has been explained by a strong solvation of the protonated azetidine ring preventing binding to the receptor [70]. The protonation also explains why the azetidine derivative is inactive towards the KB cell line [70]. Replacement of the oxygen with sulfur (**62**) affords an analogue with fifty times smaller affinity for the binding site and negligible cytotoxicity [71]. The poor affinity has been explained by (1) sulfur is a poor hydrogen bond acceptor and (2) the longer C-S bond prevents the molecule from fitting into the binding site [70].

The importance of the oxetane ring is debated. Previously it was considered essential [21]. Opening of the oxetane ring significantly reduced the effect in some studies [68]. By opening the oxetane ring, oxygen might appear outside the region, enabling hydrogen bond to Thr276. Three seco derivatives (Figure 18, **63**–**65**) only possess poor activity [72]. All these analogues, however, are not acetylated at O-4 and all have wrong stereochemistry at C-5 [69,72]. Other analogues such as (**66**) have a higher affinity for tubulin but a much lower cytotoxicity [65]

Better model compounds have been prepared and tested in silico. According to this study, compound **67** (Figure 19) would be a poor inhibitor of tubule dissembling but compounds **68** and **69** might be interesting [70]. In vitro studies of these compounds would be interesting.

The O-7 and O-10 group can be methylated without loss of activity (Cabazitaxel, Figure 20, **70**). Cabazitaxel has been approved as a drug for treatment of some cancer forms resistant to paclitaxel.

In a number of Cabazitaxel analogues with replacement of 3′-phenyl with isopropenyl, isopropyl and ethyl afford agents equipotent to or in some assays even more potent than Cabazitaxel [73]. A similar observation has been recorded for docetaxel (**39**), in which the 3′-phenyl group has been replaced with t-butyl [70]. Replacement of the O-2-benzoyl group with a benzoyl group substituted in the meta position with an azide or methoxy group has no advantage or disadvantage compared to the unsubstituted benzoate group [73]. Analogues in which O-2 has been acylated with small heteroaromatic acids like thiophen-2-carboxylic acid, furane-2-carboxylic acid, pyridine-carboxylic acid- and pyridince-3-cartboxylic acid have been patented [56,57]. All these agents show activity comparable to that of paclitaxel. Replacement of the C-7 and C-10 methoxy group with methylthiomethoxy groups similarly afforded compounds with activity comparable to that of Cabazitaxel [73]. Introduction of a methylsofoxide group severely reduces the activity (Figure 20) [73].

A surprising observation has been made for two seco analogues (Figure 21, **79** and **80**). Whereas the seco analogue (**79**) with isobutyl at C-3′ only drops to an activity one fourth of docetaxel (**39**), the analogue with phenyl at C-3′ (**80**) drops more than two orders of magnitude in activity [70]

Compound **80** was found to be more toxic than paclitaxel (**1**) in βIII-tubulin overexpressing A2780TC3 cells, but less active towards paclitaxel-sensitive cells [74]. The increased flexibility of the molecule was suggested to allow a conformation with higher affinity for the βIII-tubulin [75]. Analogues in which a linker between O-7 and O-9 has been established show high activity (Figure 22) [74].

Mammalian tubulin possesses Lys19, Val23, Asp26, His227 and Phe270, whereas yeast tubulin possesses Ala19, Thr23, Gly26 Asn227 and Tyr270. If Lys19 replaces Ala19 and His227 replaces Asn227 in mammalian β-tubulin, no major difference for the affinity of paclitaxel is observed. However, mutants possessing Thr23, Gly26 and Tyr270 have no affinity for paclitaxel [67]. Phe270Tyr mutation places a hydroxy group into a lipophilic pocket, causing steric as well as electronic conflicts for the benzamide group. Val23Thr mutation replaces a methyl group with a methyl group causing a less favorable environment for the 3′-benzamide group [67].

Different conformations of paclitaxel in the binding site have been discussed [76]. The discussion concerns a β-tubulin with Asp26, His227 and Pro358. However, human, bovine and pork β-tubulin have Asp25, Leu227 and Ile358 (RCSB PDB 1JFF, 6QVE and 5SYF). The differences between the two conformers are the torsion angles in the 3-phenylpropanoic moiety. Using REDOR NMR, the distance between groups in paclitaxel bound to β-tubulin has been measured. Based on these measurements, the below bridged paclitaxel analogues have been prepared, in which the 3′-phenyl group is linked to O-4 named britaxel-6 (Figure 22, **81**), -7 (**82**), -8 (**83**) and -5 (**84**). The working hypothesis is that the bridge forces the 13-O acyl group into the conformation found in the binding site. All britaxels were more than an order of magnitude more potent than paclitaxel. Britaxel-5 (**81**) was more than two orders of magnitude more potent towards the paclitaxel-resistant cell line 1A9-PTX10 [77,78]. All the compounds were included in a patent [79].

Another pharmacophore model pays less attention to the 2′-hydroxy group. Instead, the importance of the 15-dimethyl-1-hydroxy group moiety and the oxetane ring oxygen is emphasized [80].

Advantage of the flexibility of the substituent at O-7 was taken to prepare fluorescent derivatives of paclitaxel, Flutax 1 (Figure 23, **84**) and Flutax 2 (**85**). Both compounds have a high affinity for microtubule. Flutax 1 and 2 were used for in depth studies of the kinetic of the binding to the microtubules [81].

### 2.7. Clinical Trials of Paclitaxel and Analogues

#### 2.7.1. Clinical Use of Paclitaxel

Clinical trials with paclitaxel were initiated in 1983. Phase 1 trials revealed problems with hypersensitivity reactions primarily to the cosolvent. Poor aqueous solubility paclitaxel demands the addition of cosolvents such as Cremophor (polyethylated castor oil) and ethanol for emulsification (Taxol^®®^) [82]. To reduce allergic side effects, patients are treated with antihistamines, corticosteroids and an H2 antagonist [7]. Dose-limiting toxicity was dominated by neuropathies mainly observed as partial paralysis and numbness of extremities. The hematologic toxicology appeared to be reversible. Based on satisfactory results from clinical trials, the FDA in December 1992 approved paclitaxel for refractory ovarian cancer [68,83]. The introduction of Taxol for treatment of cancer was an incredible success. The European Medicines Agency approved paclitaxel in 1998. From 1992 to 2000, the sale raised from 20 Different conformations of paclitaxel in the binding site have been 0 million USD to 1600 million USD [10]. Since 2016, the sale has decreased because of better pharmaceutical formulations and other paclitaxel analogues with improved properties. After the loss of exclusivity, the price of a dose dropped from USD 986/dose to USSD 150/dose [6]. Consequently, 1.6 million doses were given to patients in 2016.

Resistance to paclitaxel as well as docetaxel can be caused by multidrug resistance by expression of ATP-dependent efflux pump P-glycoprotein, for which both drugs have high affinity [7]. Over expression of β-tubulin will also decrease the effects of the taxanes [7]. There is a correlation between the androgen receptor in circulating tumor cells and clinical response in treatment of hormone refractory prostate cancer cells [7].

Paclitaxel (**1**) in humans is mainly metabolized by liver cytochrome P 450 enzymes introducing a hydroxy group into the para-position of the 3′-benzyl group and at C-6 (Figure 24) [84].

#### 2.7.2. Clinical Use of Docetazel

The FDA in 1995 approved docetaxel (**39**) for treatment of advanced breast cancer, non-small-cell lung cancer, metastatic hormone refractory prostate cancer and head and neck cancer [7,85,85]. Polysorbate and ethanol must be included in the vehicle for dissolving docetaxel [82]. The sale peaked in 2014 just before loss of exclusivity (LOE) at 300 million USD [86].

The metabolism of docetaxel is depicted in Figure 25 [84]. Attempts to prepare analogues impeding metabolism by introducing fluorine atoms in the O-substituent have been made [87].

#### 2.7.3. Clinical Use of Cabazitaxel

Cabazitaxel was approved by the FDA in 2010 and the European Medicines Agency in 2013 for treatment of metastatic hormone-resistant prostate cancer in combination with prednisone. The drug is used to treat patients who do not respond to docetaxel. The two methoxy groups at O-7 and O-10 make cabazipaxel a poor substrate for ATP-dependent efflux pump P-glycoprotein [7,88]. In 2020, the sale of cabazitaxel reached €536 million [89].

#### 2.7.4. Clinical Use of Nanoparticle Formulations of Paclitaxel

The hypersensitivity induced by paclitaxel dissolved in Cremophor was removed using nanoparticle albumin-bound paclitaxel (NAB-paclitaxel, Abraxane). Abraxane consists of six or seven paclitaxel molecules bound non-covalent to an albumin molecule. This complex further aggregates to form nanoparticles of approximately 130 nm in diameter [10]. Abraxane was approved by FDA in 2005 and by the European Medicines Agency 2008 for treatment of refractory metastatic or relapsed breast cancer. The use of albumin-bound nanoparticles enables preparation of solution with a ten-times higher concentration and consequently faster infusion of the drug without addition of Cremophor [7,82]. The ability of Abraxane to eliminate breast cancer stem cells makes the drug more efficient for treatment of metastatic cancer [90] and the sale of Abraxane has increased with a rate indicating that it soon will exceed the sale of Taxol [10]. In the first six months of 2021, BMS sold Abraxane for USD 610 million [91].

Other nanoparticle and liposomal formulations of paclitaxel have been developed and some approved in China (Lipusu), South Korea (Cynviloq), Russia (Paclical), PICN (India), Nanoxel (India) and DHP-107 (South Korea) [6,10]. A series of other advanced formulations are in clinical trials and may afford more efficient formulation of paclitaxel [10].

## 3. Microtubule-Targeting Compounds

The microtubule is the target for chemotherapeutics, like the microtubule-destabilizing compounds and the microtubule-stabilizing compounds. Both kinds of chemotherapeutics have revolutionized cancer treatment.

### 3.1. Microtubule-Destabilizing Compounds

Four binding sites have been found for microtubule-destabilizing agents: (1) the vinca site, (2) the colchicine site, (3) the maytansine site and (4) the pironetin site [60]. The vinca alkaloids bind to the plus end of β-tubulin. Eribulin binds also to the plus end but only half overlapping with the vinca alkaloids. The colchicine-binding site is located on the β-tubulin at the intradimer interface of α- and β-tubulin. The maytansine-binding site is close to the vinca site but not overlapping [60].

In the 1960s, the target for colchicine (Figure 26, **93**) was identified as the tubulin, which is the building block in microtubules [92]. Binding of colchicine to the tubule destabilizes the microtubule [92]. Colchicine is the major alkaloid in *Cochicum autumnale* L. (Colchicaceae). Herbs containing colchicine have for centuries been used in traditional medicine for treatment of gout. Colchicine is an efficient drug. However, the toxicity prevents it from being approved [22]. Comparison of the structures of colchicine and compound **94** and combretastatin A-4 (CA-4) (**95**) (Figure 26) reveals the pharmacophore affording affinity for the binding domain [21]. Several other microtubule-destabilizing compounds have been identified. Compounds with affinity for the colchicine-binding site are shown in Figure 25. Podophyllotoxin (**97**) has been isolated from roots or rhizomes from *Podophyllum* species (Berberaceae). The resins were originally used as purgatives, but the discovery of the antimitotic effects made the product interesting as a chemotherapeutic [22]. The toxicity of the natural product, however, prevents its use as a drug, but the semisynthetic etoposide (**98**) and the water-soluble prodrug etopophos (**99**) is used as a drug for treatment of small lung cancer, testicular cancer and lymphoma [22]. It is to be noted that the clinical effect of etoposide is caused by inhibition of topoisomerase II [93,94].

Besides the agents with affinity for the colchicine-binding site, a group of microtubule-destabilizing compounds with affinity for the vinca domain have been characterized (Figure 27). The vinca alkaloids (**101**,**102**) have significantly improved life expectancy for patients suffering from childhood acute lymphoblastic leukemia and other cancer diseases [95].

Dolastatin 10 (**103**) was isolated from and Indian sea hare *Dolabella auricularia*. The compound failed as a drug for prostate cancer and metastatic melanoma in clinical trial II [21]. Dolastatin binds to a peptide-binding site near the vinca domain [21].

Halichondrin B (Figure 28, **105**) is a macrolide polyketide isolated from the marine sponges *Halichondria okadai.* Halichondrin B binds with affinity to the vinca-binding site [21,96]. Like paclitaxel, nature could not provide amounts of compound needed for sustainable drug production. Instead, a simplified analogue, eribulin (**106**), was synthesized [21,97]. A structure activity relationship study revealed the pharmacophore affording the affinity for microtubules resides in the right moiety of the molecule [88]. Eribulin has been approved for treatment of metastatic breast cancer for patients who previously received two chemotherapeutic regiments [88]. A sustainable supply of paclitaxel was achieved partly by developing appropriate cell cultures and semi-synthesis from more easily available starting materials. In the case of halichondrin B, a sustainable supply was obtained by simplification of the molecule to give an analogue (eribulin **106**), which in an economically feasible way could be synthesized in amounts meeting the demand [98].

Maytansine 1 (**107**) conjugated via a linker to an HER-2 targeted antibody has been approved for treatment of breast cancer overexpressing the HER-2 gene [21,99].

### 3.2. Microtubule-Stabilizing Compounds

Only two binding sites have been found for microtubule stabilizers 1) the taxane site and the laulimalide/peloruside site [60]. The taxane-binding site is located on the interior of the microtubule. The other binding site is on the exterior of the microtubule [60]. The taccalonolides, zampanolides and cyclostreptin bind covalently to the binding site [60].

Many natural products have been found to stabilize microtubules (Figure 29). Peloruside A (**108**) together with three related polyketide macrolides were isolated from the marine sponge *Mycale hentscheli* collected in southern New Zealand [100]. The binding site of peloruside A involves Phe294, Tyr310, Arg306 and Tyr340 of the exterior site on β-tubulin [100]. The development of peluroside into a drug has been delayed due to a supply issue [101]. Laulimalide (**109**) was isolated from the marine sponge *Cacospongia mycofijiensis* found in the Pacific Ocean [102]. Both compounds are microtubule stabilizers binding to the same binding site [100].

The taccalonolides (Figure 30) have been isolated from *Tacca chantrieri* (André), Dioscoraceae, in order to characterize the principle causing bundling of interphase microtubules [103]. Other taccalonolides have been isolated from other *Tacca* species. Taccalonolides A (**110**) and E (**112**) shown in Figure 29 are examples of the more than 20 known steroids all possessing the same carbon skeleton. Taccalonolides B (**111**) and N (**113**) are semisynthetic analogues. Characteristics for the taccalonolides are the C-2-C-3 epoxy group and the C-23-C-24 enol γ-lactone group. Taccalonolides A, B, E and N cause bundling of interphase microtubules and mitotic arrest followed by apoptosis [103]. The taccalonolides also provoke apoptosis in cells with mutated paclitaxel-binding sites and expression of P-glycoprotein (Pgp). Even though the taccalonolides show poor in vitro activity, they have high in vivo potency. However, studies have revealed that they, in contrast to taccalonolides AF (110) and AJ (**111**), do not bind to microtubules. In addition, taccalonolide AF and AJ are several orders of magnitude more potent than taccalonolides A and B [104]. This has led to the hypothesis that enol esters **110** and **111** are prodrugs of epoxides **114** and **115**, respectively. Taccalonolide AF has been isolated from *Tacca plantaginea* (Hance) [104]. However, the compound was only present in low abundance. Consequently, semisynthesis was performed using taccalonolide A as starting material. Synthesis provided enough compound to establish microtubule-stabilization activity. Taccalonolide AJ is a semisynthetic product prepared by epoxidation of taccalonolide B [104]. In early publications, the C-22-C-23 epoxide was α-disposed; later publications suggest a β-disposed epoxide [101,105].

Cyclostreptin (Figure 31, **116**) isolated from a culture of *Streptomyces* sp. 9885 can displace Flutax 2 (**86**) from microtubules. In contrast, Flutax 2 cannot displace cyclostreptin [106]. Mass spectrometric analysis of fragments of microtubules reveals that the fragment containing amino acid residues 219 to 243 gains an *m/z* value of 133.2 after incubation of the microtubule with cyclostreptin. Since the fragment has a charge of 3 (*z* = 3), this corresponds to an increased molecular weight of 400.2, corresponding to an addition of cyclostreptin to the fragment (Figure 31). It is not stated if the addition is an amide formation or a hetero Michael addition. Both Thr220 and Asn228 are suggested to be the nucleophile reagent attacking cyclostreptin [106].

Zampanolide (Figure 32, **118**) was isolated from the marine sponge *Fasciospongia rimasa* and from *Cacospongia myco**fi**jiensis*, although only in small amounts. Total syntheses of the compound have been developed [107,108]. Dactylolide (**119**) was isolated from a *Dactylospongia* sp. sponge [108]. Very varying measurements of optical rotations maybe because of varying degree of enolization have complicated the establishment of the absolute configuration [107]. The missing aminal chain reduces the cytotoxicity of dactyloide by two orders of magnitude compared with zampanolide.

Both zampanolide and dactyloide bind to the binding site of paclitaxel on the microtubule. An interesting feature of zampanolide is the toxicity towards multi-drug-resistant cancer cell lines expressing P-gp pump [107]. The failure of P-gp to remove zampanolide might be caused by formation of a covalent bond to the microtubules. A hetero-Michael reaction between the heterocyclic nitrogen atom of His230 to give **120** or the amide of Asn229 to give **121** and C-9 has been suggested as the reaction forming the covalent bond zampanolide (Figure 32) [107,108]. Dactyloide may react in a similar way. Other possibilities, however, are mentioned. None of the suggested nitrogen atoms, however, are likely to undergo a hetero-Michael reaction.

## 4. Conclusions

The development of procedures for sustainable large-scale production of paclitaxel enabling sufficient supply of the drug caused a revolution in the treatment of previously fatal cancer diseases, causing and an increase life expectancy. The annual sale of the drug exceeded 1600 million USD in 2016, meaning that more than 1.6 million doses have been given to patients. A problem with paclitaxel is the poor aqueous solubility, forcing addition of cosolvents such as Cremophor, which might induce allergic reactions. In addition, resistance to the drug has developed. Doxtaxel has increased water solubility and cabazitaxel is a poor substrate for the P-glycoprotein that removes paclitaxel from the cancer cells. A new nanoparticle formulation of paclitaxel Abraxane has to some extent solved the problem, with solubility enabling higher doses of the drug. Despite the optimization of the taxanes, new drugs for treatment of cancer patients are needed since many of the present drugs increase the progression-free survival period of patients, but do not eliminate the cancer. Some hope might be found in the fact that investigation of new sources of the drug in, e.g., marine organisms and fungi have expanded the possibilities for finding completely novel structures. Some examples of such drugs are the macrolides eribulin and maytansine mentioned in Section 3 of this review. Eribulin has been approved for treatment of metastatic breast cancer in patients who have received two to or more other chemotherapeutic regiments. Maytansine, however, is not used as a drug but as a warhead after coupling with an antibody. The use of antibodies to target chemotherapeutics is a promising new development. Even though phytochemical investigations have revealed new compounds with surprising properties like the taccalonolides, natural products obtained from marine organisms or fungi might give novel compounds with beneficial effects.

## Figures and Tables

**Figure 1 molecules-27-05648-f001:**
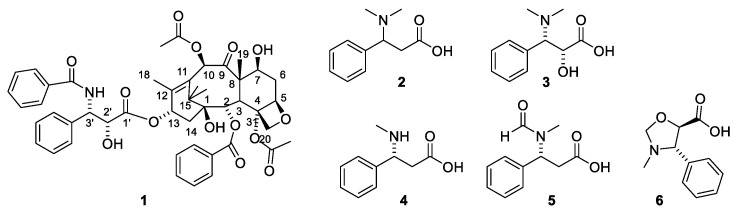
Paclitaxel (**1**) and phenylisoserine (**2**–**6**) analogues esterified with diterpenes.

**Figure 2 molecules-27-05648-f002:**
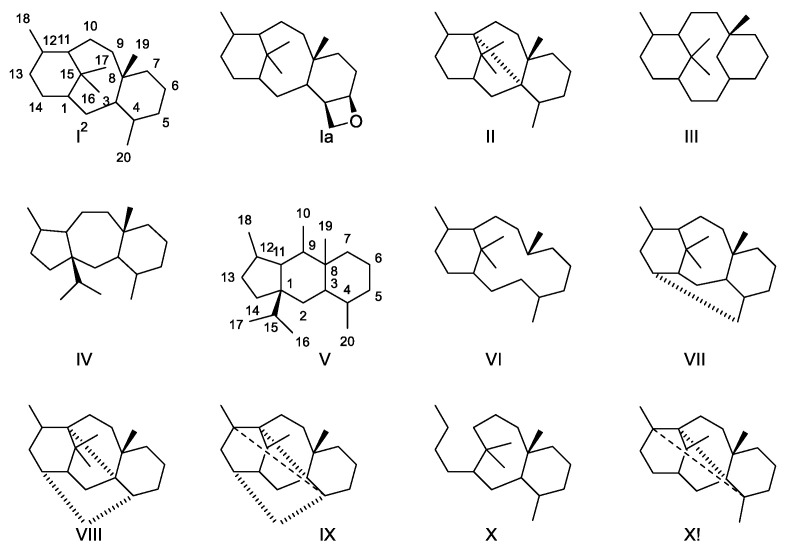
Skeletons found in diterpenes isolated from species belonging to *Taxus* [14]. In a nomenclature system, only group I (and Ia) possessing three rings (6/8/6) is named taxanes, group II (6/5/5/6) is the 3,11-cyclotaxane, group III (6/10/6)2(3-20)abotaxane, group IV (5/7/6) 11(15-1)abotaxane, group V 5/6/6 (11(15-1),11(10-9)diabotaxane, group VI 6/12 (3,8)secotaxane, group VII 6,8,6,6 (14-20)cyclotaxane, group VIII 6/5/5/6 (3,11:12,20)dicyclotaxane, group IX 5/5/4/6/6/6 (3,11:4,12:14,20)tricyclotaxane, group X 8/6 (11,12)secotaxane and group XI 6/5/5/4/6 (3,11:4,12)dicyclotaxanes [13,23].

**Figure 3 molecules-27-05648-f003:**
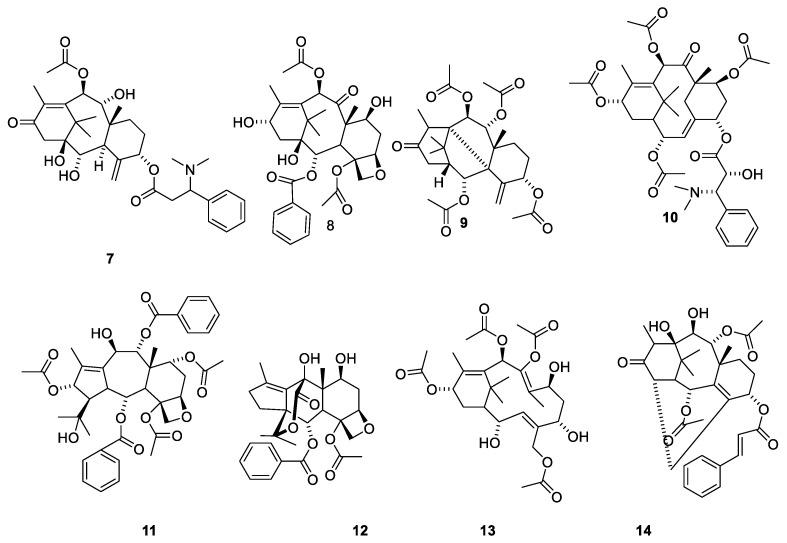
Representative compounds belonging to the 11 groups of diterpenes isolated from *Taxus* species [14,23]. Taxine B (**7**), baccatin III (**8**). Taxinine L (**9**), α,7β,10β,13α-Tetraacetoxy-5α-(2′ R,3′ S)–*N*,*N*-dimethyl-3′-phenylisoseryloxy]-2(3→20)abeotaxa-4(20),11-dien-9-one (**10**), taxchinine I (**11**), tasumatrol j (**12**), Tasumatrol M (**13**), 2α,9α-Diacetoxy-5α-cinnamoyloxy-10β,11β- ihydroxy-14β,20-cyclotax-3-en-13-one (**14**) [23].

**Figure 4 molecules-27-05648-f004:**
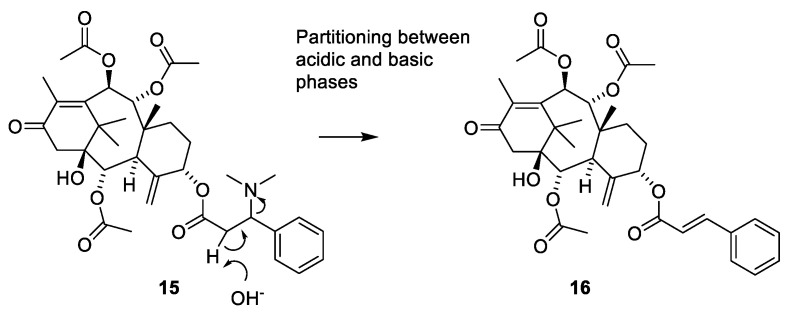
Structure elucidation of taxine I. During isolation, the dimethylamino group in diacetyltaxine B (**15**) is eliminated to give **16**. Solvation catalyzed with sodium methanolate affords the tetraol **17** (Figure 5). Hydrogenation afforded **18** [36]. Application of ^1^H NMR spectroscopy confirmed the structure and enabled establishment of stereochemistry [35].

**Figure 5 molecules-27-05648-f005:**
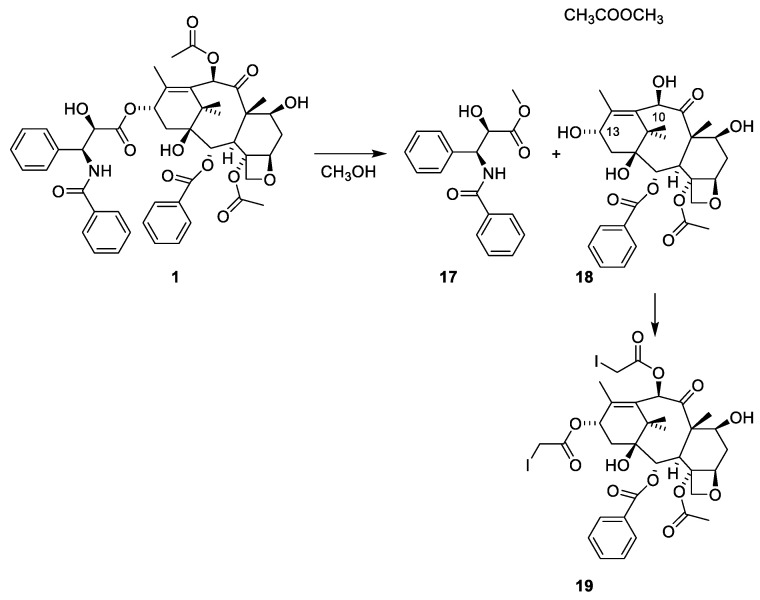
Methanolysis of paclitaxel (**1**) to give the tetraol (**18**) and the benzoylated phenylisoserine ester **17**. The structure of **18** was solved with an X-ray analysis of the diiodoacetyl ester **19** [38].

**Figure 6 molecules-27-05648-f006:**
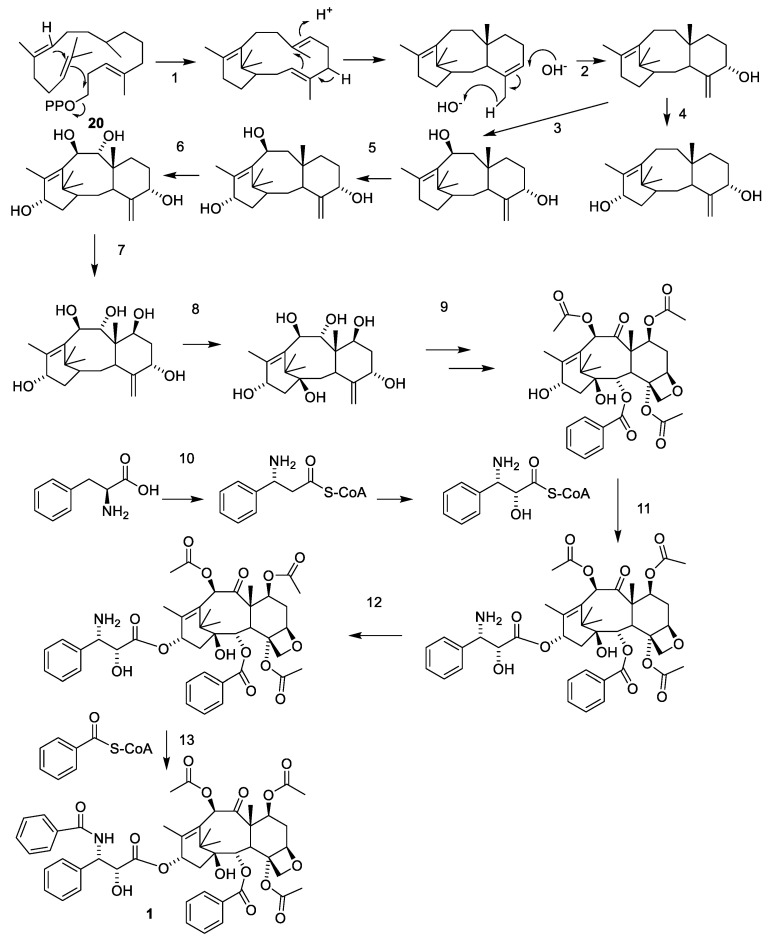
Biosynthesis of paclitaxel (**1**) starting form geranylgeranyl pyrophosphate (**20**). The sequence of oxidation begins with C-5 then C-10, C-13, C-9, C-7 and finally C-2. The oxetane group is introduced by epoxidation of the C-4-C-20 double bond. The oxidation of C-9 occurs at the same time as the formation of the oxetane group. The O-13 side chain is introduced as isophenylalanine, which after esterification is oxygenated at C-2 and *N*-benzoylated. Enzymes: taxadiene synthase (**1**), taxadiene 5α-hydroxylase (**2**), taxane 10β-hydroxylase (**3**), taxane 13α-hydroxylase (**5**), taxoid 14β-hydroxylase (**6**), a complex of some characterized enzymes such as taxoid 7β-hydroxylase (**7**), taxoid 2β-hydroxylase (**8**), a class of enzymes including a taxoid 2α-benzoyltransferase, taxoid 10β-acetyltransferase and taxoid C9 hydroxyl oxidase (**9**), taxoid C13 phenylpropanoid side chain CoA acyltransferase (**11**), oxidase oxidating C-2′ and taxoid C13 side chain N-benzoyltransferase (**13**). Some enzymes and the sequence of the oxidations are still debated [22,23,29,39].

**Figure 7 molecules-27-05648-f007:**
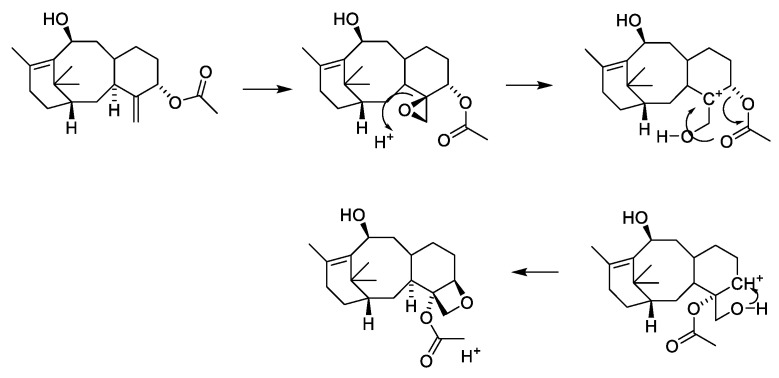
Acid-catalyzed formation of the oxetane ring [40,41].

**Figure 8 molecules-27-05648-f008:**
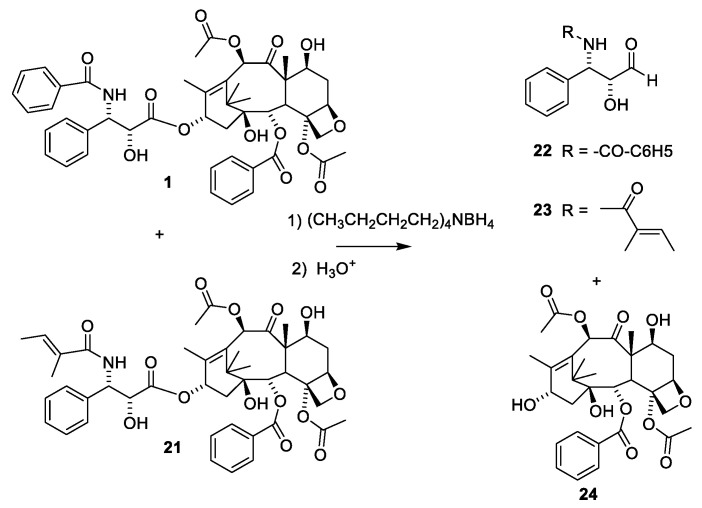
Reductive de-esterification of paclitaxel (**10**) and cephalomannine (**21**) [47].

**Figure 9 molecules-27-05648-f009:**
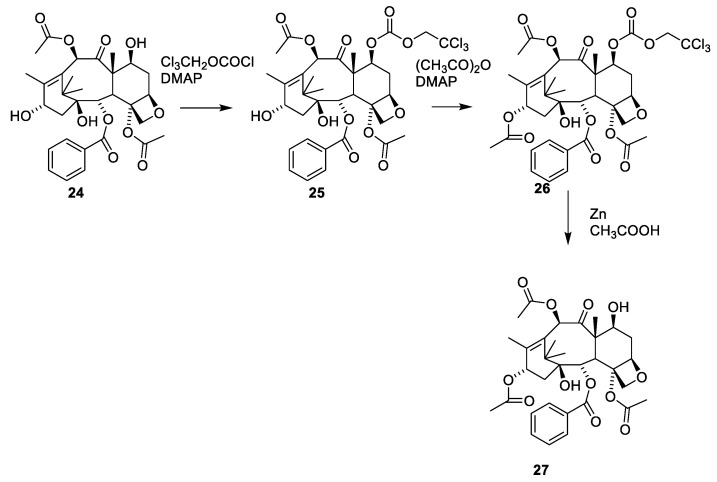
Selective acylation of O-7 and O-13 of taxanes [47]. DMAP is 4-dimethylaminopyridine.

**Figure 10 molecules-27-05648-f010:**
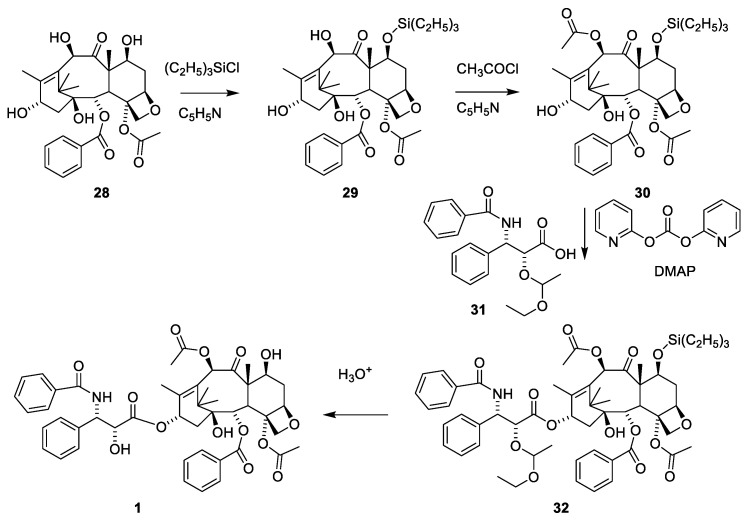
Semi-synthetic synthesis of paclitaxel (**1**) from the easy available 10-deacetylbaccatin III (**28**) to give paclitaxel in 4 steps [50]. DMAP dimethylaminopyridine.

**Figure 11 molecules-27-05648-f011:**
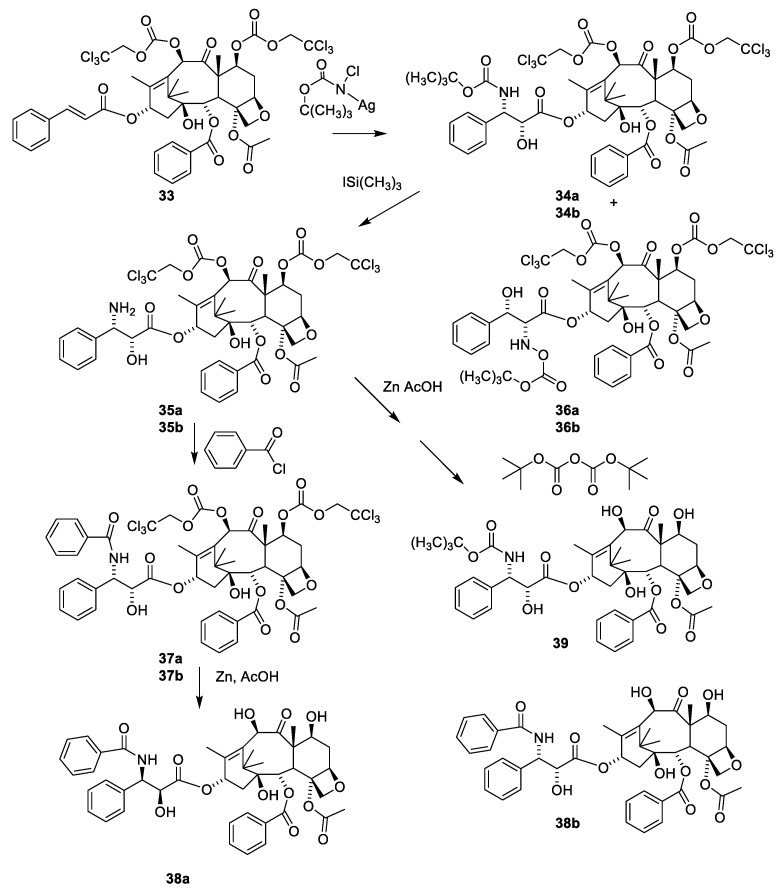
Semi-synthetic preparation of a mixture of regio- and stereoisomers of 10-deactyltaxol (**38b**). 10-Deacetylbaccatin was protected using Troc at the reactive 7- and 10-hydroxy groups followed by cinnamoylation with ciannamoyl chloride to give **33**. Sharpless oxyamination of **33** affords a mixture of all four possible regio- and stereoisomers **34a**, **34b**, **36a** and **36b**. Deprotection of the amino group affords the free amine (e.g., **35a** and **35b**), which after bezoylation give benzamide (**37a** and **37b**). Deprotection of the 7- and 10-hydroxy groups afforded 10-deacetyl paclitaxel (**38b**) [51]. Selective 10-acetylation may be obtained as suggested in Figure 9. Removal of the Troc groups using zinc in acetic acid in methanol from the mixture of the four isomers **34a**/**34b** and **36a**/**36b** afforded four products including docetaxel (**39**) [13].

**Figure 12 molecules-27-05648-f012:**
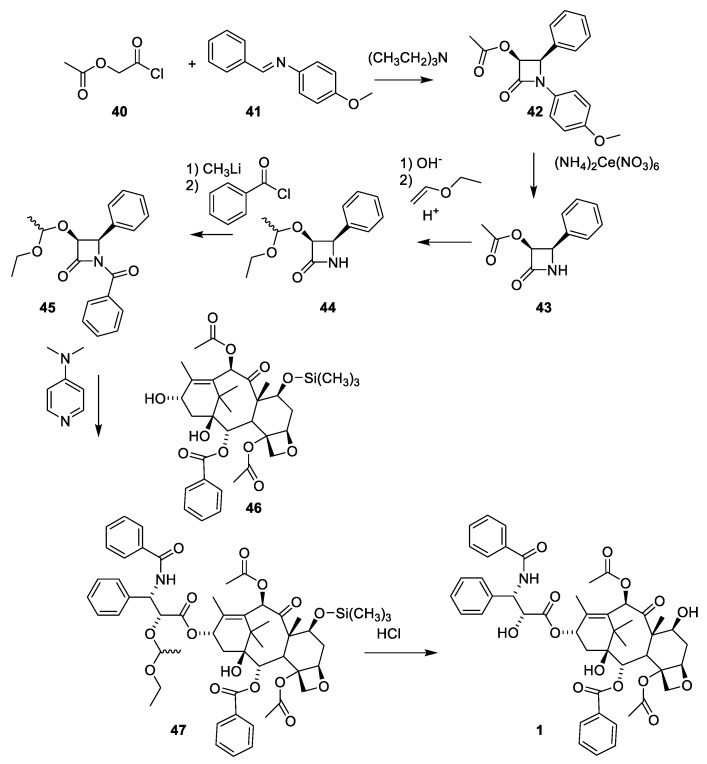
Preparation of paclitaxel (**1**) by semi-synthesis from trimethylsilylatedbaccatin III (**46**) taking advantage of the acylating properties of β-lactames. The stereochemistry of the intermediates is complicated by the chiral-protecting group [52]. The prepared paclitaxel is contaminated with the isomer with opposite stereochemistry in the isophenylserine group.

**Figure 13 molecules-27-05648-f013:**
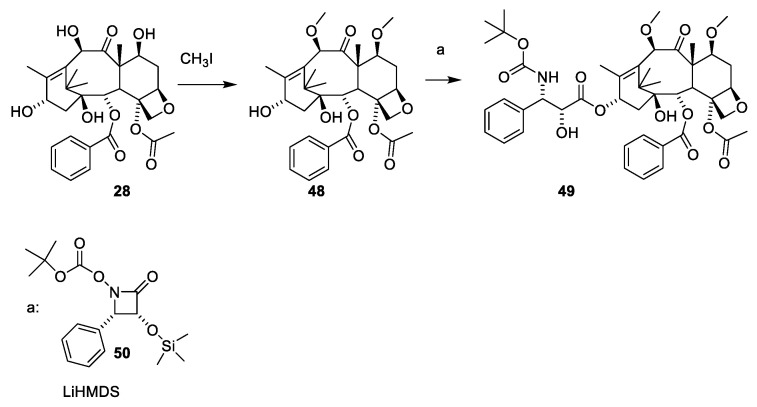
Selective alkylation of deacetylbaccatin III (**28**) to give dimethoxy deacteylbaccatin III (**48**) [53,58]. Reaction of **48** with the β-lactam **50** affords cabazitaxel (**49**). LiHMDS: lithium hexamethyldisilazide.

**Figure 14 molecules-27-05648-f014:**
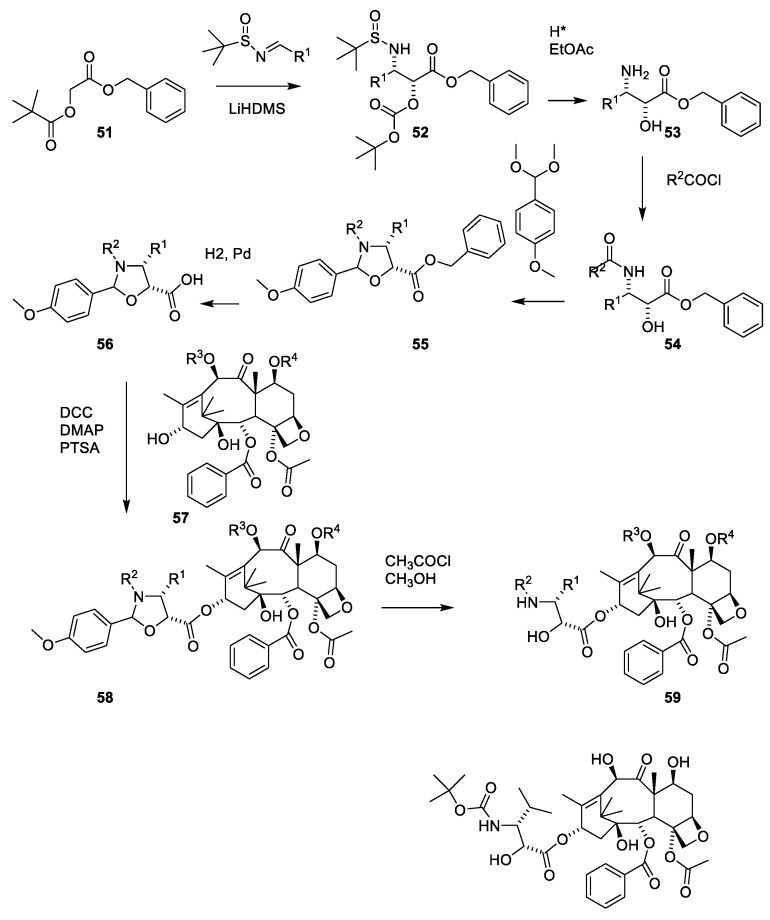
Preparation and use of oxazolidine-5-carboxylic acid (**56**) for esterifying O-13 of masked deacetylbaccatin III (**57**) [59]. LiHDMS: lithium hexamethyldisilazide, DCC: dicyclohexylcarbodiimide, DMAP: 4-dimethylaminopyridine, PTSA: paratoluenesulfonic acid [59].

**Figure 15 molecules-27-05648-f015:**
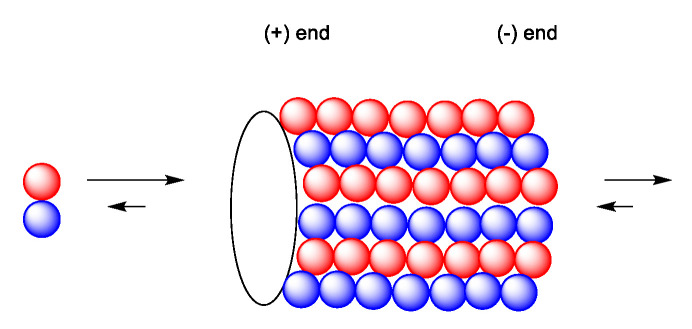
Dynamic state of the microtubule continuously prolonged by complexing or shortened by and dissociating heterodimer tubules from the microtubule [5,60].

**Figure 16 molecules-27-05648-f016:**
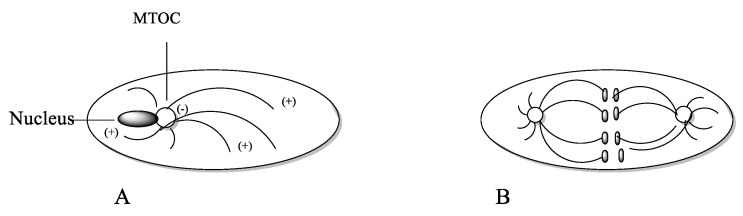
(**A**) Resting cell: Orientation of microtubule [60]. (**B**) Anaphase. The duplicated chromosomes attached to the microtubules are moved towards the spindle poles [5].

**Figure 17 molecules-27-05648-f017:**
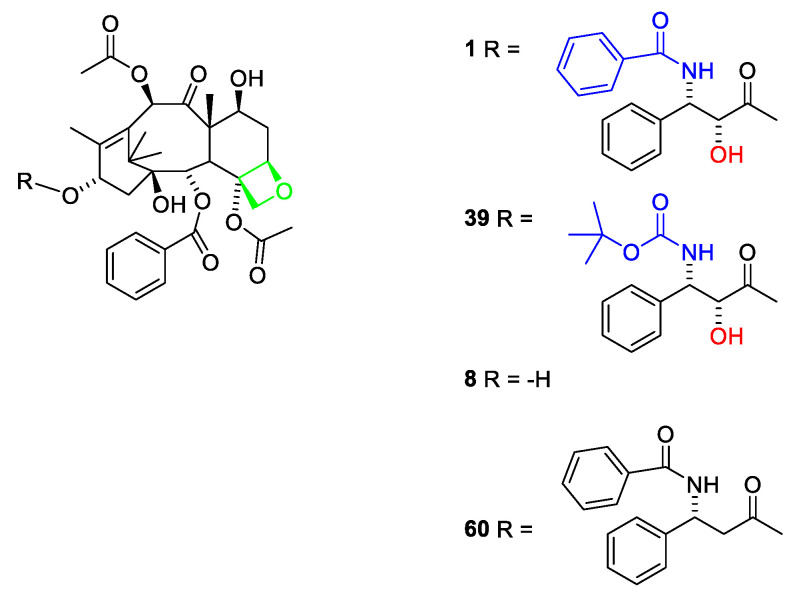
Structure activity relationships for the substituent at O-13. The hydroxy group at C-2′ (red) is critical for affinity to microtubules. The C-3′ acylamino group (blue) is important for the correct conformation of the diterpene nucleus [66,67].

**Figure 18 molecules-27-05648-f018:**
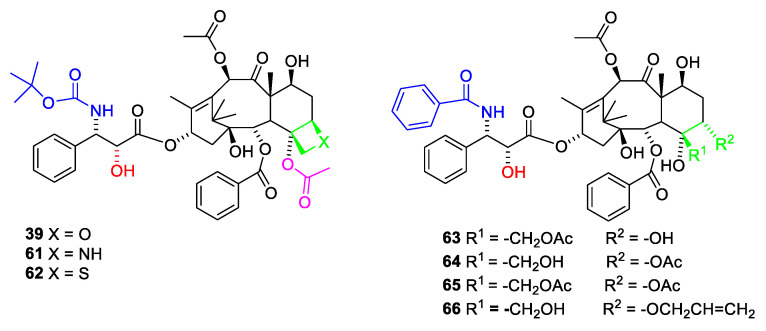
Docetaxel (**39**) analogues in which the oxetane oxygen has been replaced with nitrogen (**61**) of sulfur (**62**) [69]. Paclitaxel analogues in which the oxetane ring (green) has been opened (**63**–**66) [65,72]**. The 4-acetoxy group is magenta.

**Figure 19 molecules-27-05648-f019:**
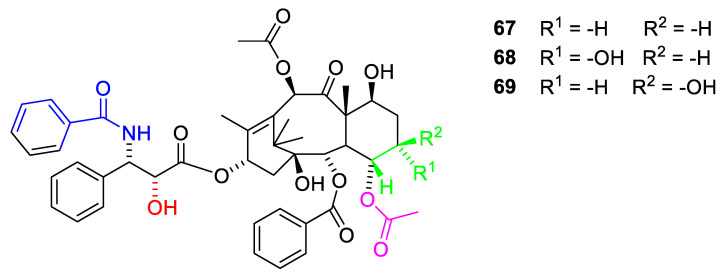
Paclitaxel analogues with no oxetane ring [70].

**Figure 20 molecules-27-05648-f020:**
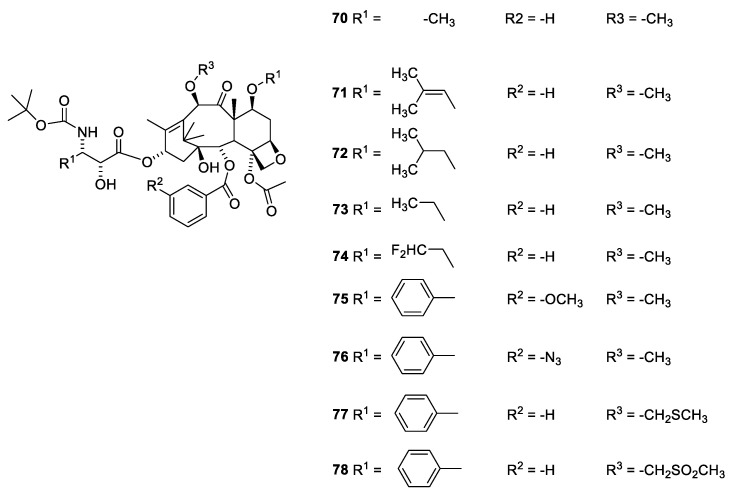
Cabazitaxel (**70**) and analogues in which the C-3′ and the O-9 and O-10 substituents have been varied (**71**–**78**) [73].

**Figure 21 molecules-27-05648-f021:**
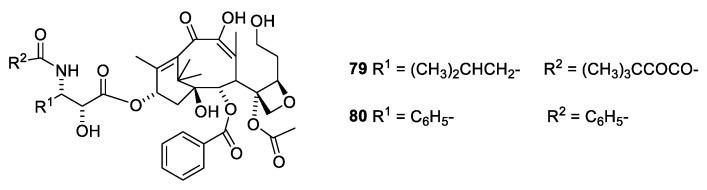
Seco analogues of paclitaxel [70].

**Figure 22 molecules-27-05648-f022:**
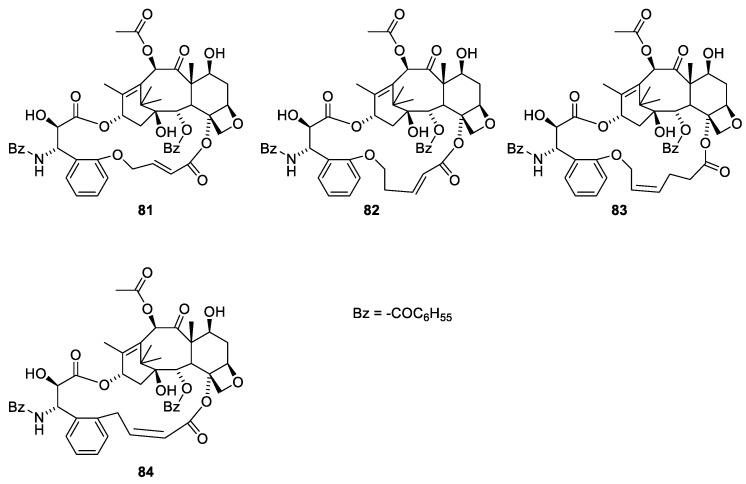
Britaxel-5 (**81**), -6 (**82**), -7 (**83**) and -5 (**84**) [77,78].

**Figure 23 molecules-27-05648-f023:**
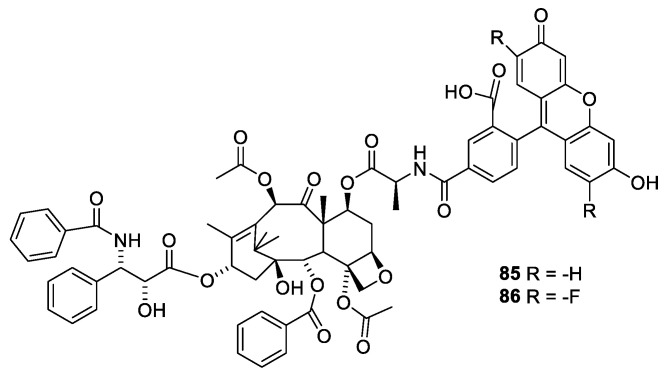
Flutax 1 (**85**), Flutax 2 (**86**) [81].

**Figure 24 molecules-27-05648-f024:**
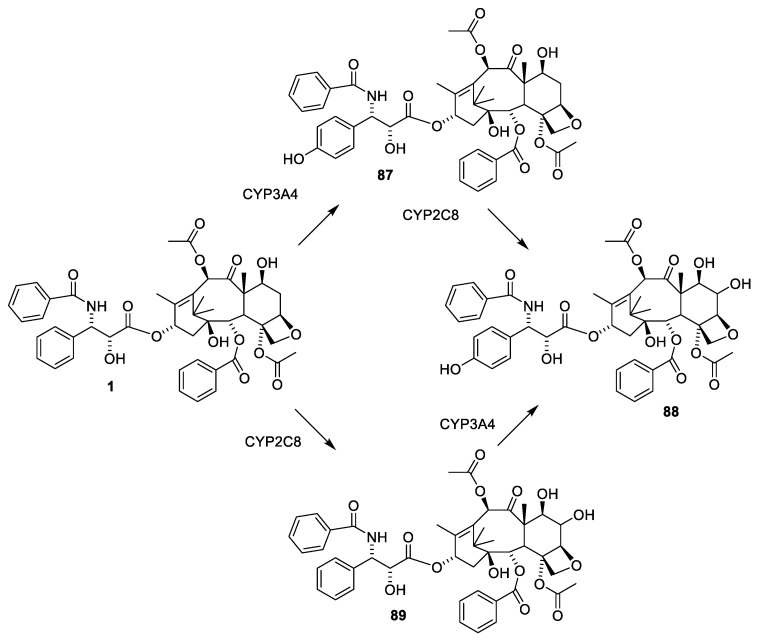
Dominant metabolism of paclitaxel by liver cytochrome P450 enzymes CYP2C8 and CYP3A4 [84].

**Figure 25 molecules-27-05648-f025:**
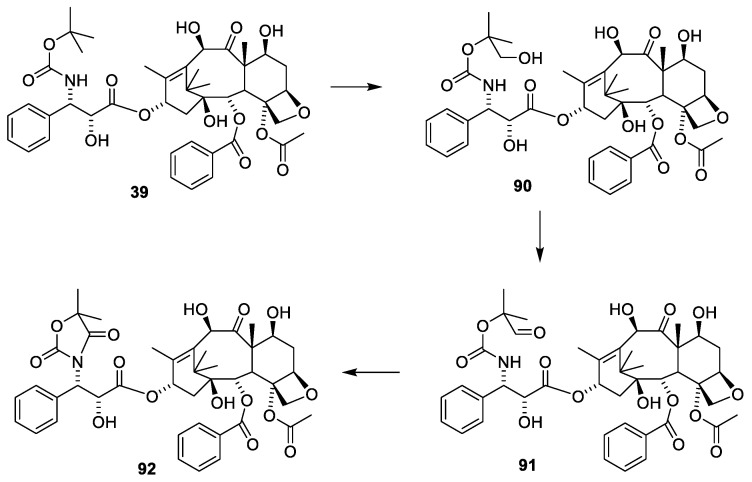
Metabolism of docetaxel (**39**) [84].

**Figure 26 molecules-27-05648-f026:**
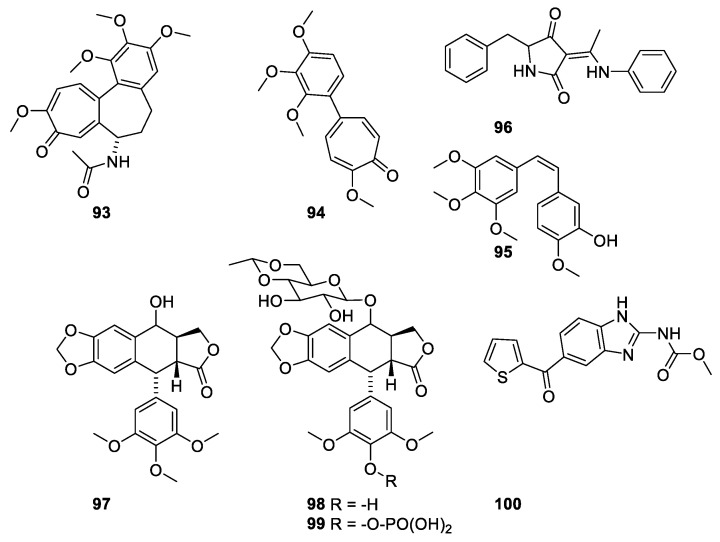
Compounds with affinity for the colchicine domain in tubulin. Colchicine (**93**), the colchicine pharmacophore 2-methoxy-5-(2′,3′,4′-trimethoxyphenyl)tropone (**94**) and combretastatin CA-4 (**95**), TN-16 (**96**), podophyllotoxin (**97**), etoposide (**98**), etopophos (**99**) and nocodazole (**100**) [92].

**Figure 27 molecules-27-05648-f027:**
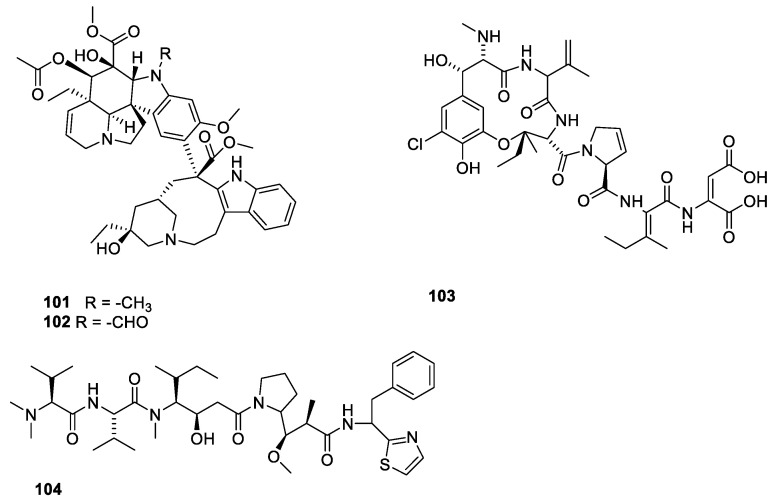
Microtubule-destabilizing agents with affinity for the vinca domain: Vinblastin (**101**), vincristine (**102**), phomopsin A (**103**), dolastatin 10 (**104**) [21,92].

**Figure 28 molecules-27-05648-f028:**
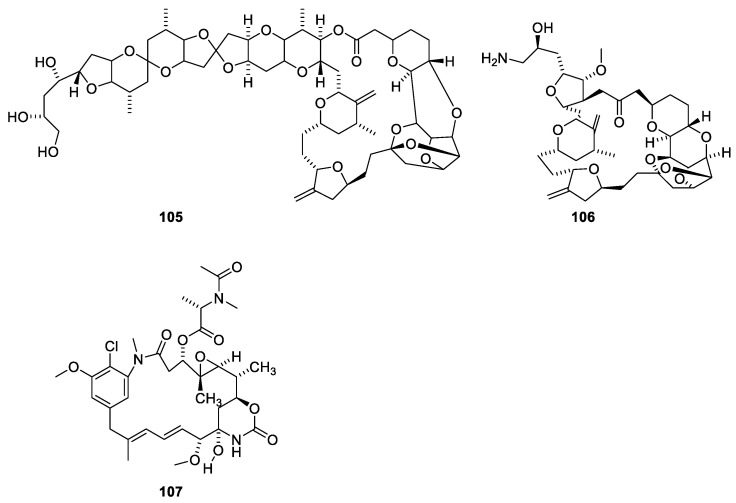
Halichondrin B (**105**) and a simplified analogue, eribulin (**106**), approved for treatment of breast cancer. Maytansine 1 (**107**) conjugated via a linker to an antibody-targeting HER-2 has been approved for treatment of breast cancer [21].

**Figure 29 molecules-27-05648-f029:**
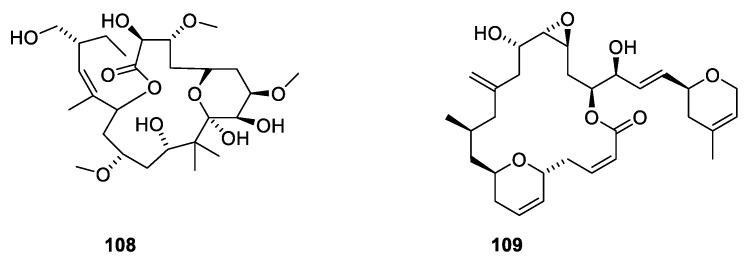
Microtubule-stabilizing agents peluroside A (**108**), laulimalide (**109**).

**Figure 30 molecules-27-05648-f030:**
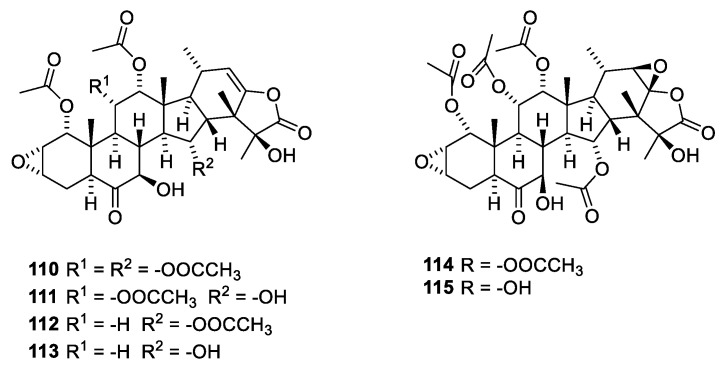
The taccanlonolides. Taccalonolides A (**110**) and E (**112**) were isolated from tubules of *Tacca chantrieri.* Taccalonolides B (**111**) and N (**113**) are semisynthetic analogues. Taccalonolides AF (**114**) and AJ (**115**) [103].

**Figure 31 molecules-27-05648-f031:**
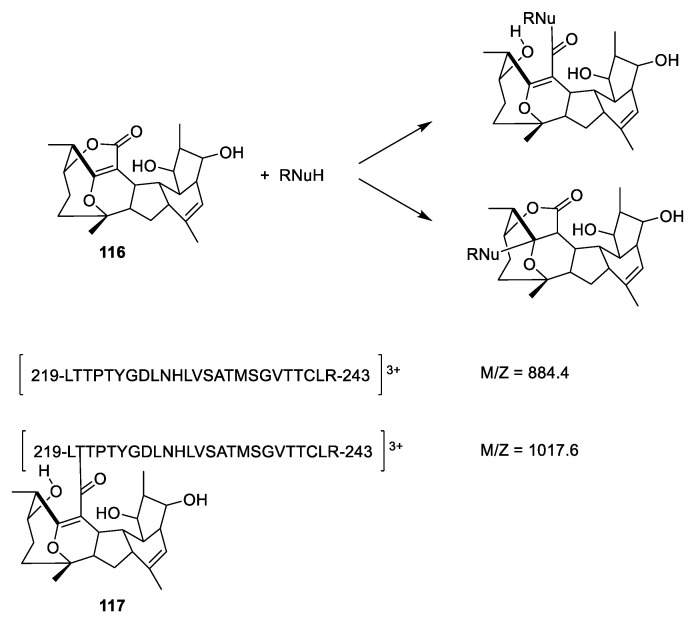
Possible paths for reacting cyclostreptin (**116**) with microtubules. Either the lactone opens forming an amide or a hetero-Michael reaction affords a 1,4-addition to the α,β-unsaturated lactone [106]. A possible reaction product (**117**) has been suggested.

**Figure 32 molecules-27-05648-f032:**
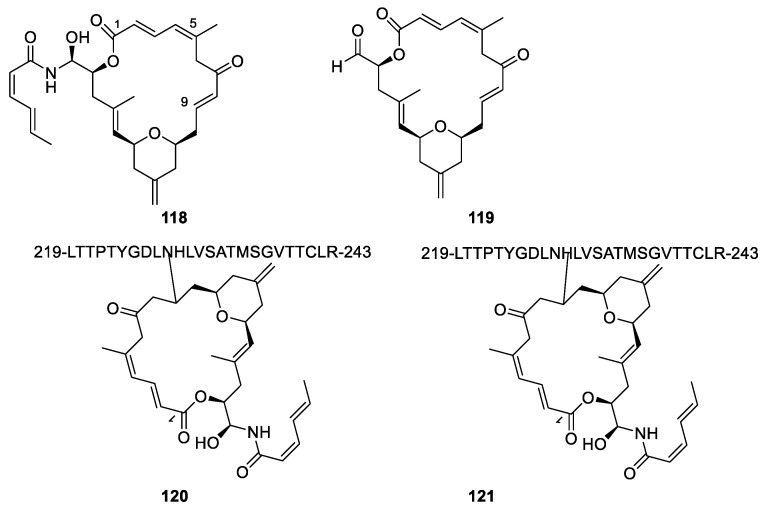
Zampanolide (**118**) and dactyloide (**119**) and suggested products (**120** and **121**) formed by reacting microtubule with zampanolide [107,108].

## Data Availability

Not applicable.

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
