# Peer review of "Drugs That Changed Society: Microtubule-Targeting Agents Belonging to Taxanoids, Macrolides and Non-Ribosomal Peptides"

_molecules, 2022, doi:10.3390/molecules27175648_

Round 1

Reviewer 1 Report

This review is a nice piece of work in the field of drug development. I congratulate the author for his tremendous effort in compiling this review. 

Author Response

Reviewer 1

This review is a nice piece of work in the field of drug development. I congratulate the author for his tremendous effort in compiling this review. 

I thank the reviewer for the kind comment.

Reviewer 2 Report

The paper is structural and good. However, I have some concerns and comments:

1) I do not agree with the suggested title, because it is too broad and gives an exaggerated impression.

2) The quality of figures 15 and 16 should be improve, and if they are not created by authors, they should indicate adapted by and give ''copyright assessment''

3) The number of figures concerning chemical compounds should be reduced 

4) I suggest to add one or two figure showing molecular mechanisms on microtubules 

5) Authors use references in the conclusion part. However, in this part, concluding remarks with some perspectives should be indicated in this part without citing references (The personal opinion of the author and his suggestions for perspectives concerning the applications of these molecules)

6) Toxicological aspects should be  highlighted to show to safety of these compounds 

7) I suggest to add another point concerning cellular and molecular impacts (cell division) with molecules targeted microtubules. 

Author Response

Reviewer 2
The paper is structural and good. However, I have some concerns and comments.:

I thank the reviewer for the kind comment.

  • I do not agree with the suggested title, because it is too broad and gives an exaggerated impression.
    I have narrowed the scope of the tile.

2) The quality of figures 15 and 16 should be improved, and if they are not created by authors, they should indicate adapted by and give ''copyright assessment''
Fig. 15 has been changed

3) The number of figures concerning chemical compounds should be reduced 
The main purpose of this manuscript is to review the chemical diversity of compounds targeting microtubules. Consequently, I find it is important that many structures are shown.

4) I suggest to add one or two figure showing molecular mechanisms on microtubules
Based on the literature find it very difficult to  make such figures.

5) Authors use references in the conclusion part. However, in this part, concluding remarks with some perspectives should be indicated in this part without citing references (The personal opinion of the author and his suggestions for perspectives concerning the applications of these molecules)
References have been removed

6) Toxicological aspects should be highlighted to show to safety of these compounds 
In-depth toxicological aspects are beyond the goal of this review

7) I suggest to add another point concerning cellular and molecular impacts (cell division) with molecules targeted microtubules. 
Apparently, new research reveals that the mechanism of action of rather is caused by inhibition of cellular trafficking of molecules and organelles than inhibition of cell division.

Reviewer 3 Report

The author conducted a study entitled "Drugs that changed society: Microtubule targeting Agents. Terpenoids, Polyketide Macrolides and non-Ribosomal Peptides"

In general the manuscript is well written and in my view it needs improvement in just a few points.

The author can insert a methodology section, in it he was put the entry mechanisms for the search and selection criteria of the articles used in the research, I believe that this can facilitate the understanding of future readers.

In my view, the study was carried out based on results from Europe, I believe that this can appear in the title of the work, in addition the introduction can be divided into paragraphs and not just one.

Some molecules are classified as diterpenes, however, they have more than 30 carbon atoms, so they would be terpenoids or triterpenes, this must be corrected. Example fig 3.

In terms of class of compounds, Taxine is an alkaloid

Remove all references from the conclusion and conclude according to the objectives of the manuscript, analyze what advances your work brings to literature and what future perspectives in the area should be made clear.

Author Response

Reviewer 3

The author conducted a study entitled "Drugs that changed society: Microtubule targeting Agents. Terpenoids, Polyketide Macrolides and non-Ribosomal Peptides"

In general the manuscript is well written and in my view it needs improvement in just a few points.

The author can insert a methodology section, in it he was put the entry mechanisms for the search and selection criteria of the articles used in the research, I believe that this can facilitate the understanding of future readers.
In the introduction I have elaborated on the purpose of this review. That is the review has focused on the chemistry of the terpenoids and mechanism of action of newer drugs. In addition, new literature has been approached. A sentence has been added.

In my view, the study was carried out based on results from Europe, I believe that this can appear in the title of the work, in addition the introduction can be divided into paragraphs and not just one.
The review has not taken reference to the situation in Europe.

Some molecules are classified as diterpenes, however, they have more than 30 carbon atoms, so they would be terpenoids or triterpenes, this must be corrected. Example fig 3.
It is true that all compounds in fig. 3 contains more carbon atoms than 15. However, all the nucleus of all compounds contains 20 carbons, some oxygenated and the alcohols esterified. Such side chains are not considered part of the nucleus and consequently, the compound is still considered a diterpene.

In terms of class of compounds, Taxine is an alkaloid. Taxine is an early term for a fraction isolated from T. brevifolia. Later studies have revealed that this fraction contained a number of compounds as described in this manuscript.

Remove all references from the conclusion and conclude according to the objectives of the manuscript, analyze what advances your work brings to literature and what future perspectives in the area should be made clear.
